# Antimicrobial Stewardship Programs in Community Health Systems Perceived by Physicians and Pharmacists: A Qualitative Study with Gap Analysis

**DOI:** 10.3390/antibiotics8040252

**Published:** 2019-12-05

**Authors:** Sohyun Park, Ji Eun Kang, Hee Jung Choi, Chung-Jong Kim, Eun Kyoung Chung, Sun Ah Kim, Sandy Jeong Rhie

**Affiliations:** 1Graduate School of Pharmaceutical Sciences, Ewha Womans University, 52 Ewhayeodae-gil, Seodaemun-gu, Seoul 03760, Korea; 2College of Pharmacy, Ewha Womans University, 52 Ewhayeodae-gil, Seodaemun-gu, Seoul 03760, Korea; jjadu@nmc.or.kr; 3Department of Pharmacy, National Medical Center, Seoul 04564, Korea; 4College of Medicine, Ewha Womans University, 52 Ewhayeodae-gil, Seodaemun-gu, Seoul 03760, Korea; heechoi@ewha.ac.kr (H.J.C.); cj.kim.id@gmail.com (C.-J.K.); 5Department of Internal Medicine, Ewha Womans University Mokdong Hospital, 1071, Anyangcheon-ro, Yangcheon-gu, Seoul 07985, Korea; 6Department of Internal Medicine, Ewha Womans University Seoul Hospital, 260, Gonghang-daero, Gangseo-gu, Seoul 07804, Korea; 7College of Pharmacy, Kyung Hee University, 26, Kyungheedae-ro, Dongdaemun-gu, Seoul 02447, Korea; cekchung@khu.ac.kr; 8Department of Pharmacy, Kyung Hee University Hospital at Gangdong, 892 Dongnam-ro, Gangdong-gu, Seoul 05278, Korea; 9Department of Pharmacy, Ewha Womans University Mokdong Hospital, 1071, Anyangcheon-ro, Yangcheon-gu, Seoul 07985, Korea; ssk320@hanmail.net

**Keywords:** antimicrobial stewardship program, pharmacist, interdisciplinary team, community health system

## Abstract

Antimicrobial stewardship program (ASP) is one of the most important strategies for managing infectious disease treatment and preventing antimicrobial resistance. The successful implementation of ASP in the community health system (CHS) has been challenging. We evaluated perceptions of current ASP, potential setbacks of ASP implementation, and future demands on ASP services among physicians and pharmacists in the CHS. The qualitative research was conducted through in-depth individual interviews and focus group discussions with 11 physicians and 11 pharmacists. In addition, a quantitative gap analysis was conducted to assess the different awareness and demands on services of ASP and preferred antimicrobial-related problems (ARP). In overall, perceptions of ASP varied by profession. The identified setbacks were unorganized institutional leadership, the undefined roles of healthcare professionals, a lack of reimbursement, the hierarchical structure of the health system, and the labor-intensive working environment of pharmacy services. Although demands for ASP improvement were similar among professionals, they had different preferences in prioritizing each service item of ASP/ARP development and the profession responsible for each service. Continuous administrative and financial investments, understanding ASP contents, ASP-specific information technology, and interdisciplinary collaboration with good communication among healthcare professions are needed to continue the progression of ASP.

## 1. Introduction

In recent years, infection prevention and control (IPC) have become a challenge and a topic of interest for medical institutions in Korea [1,2]. The Accreditation Program for Healthcare Organization has presented criteria for IPC with the aim of improving patient safety and quality of care [3]. Since 2017, the Korean Ministry of Health and Welfare (MOHW) has been providing funds to support medical quality assessment of medical institutions [4]. Among the areas evaluated for medical quality and patient safety, factors related to IPC include personnel for IPC, antimicrobial prescription rates, and participation in antimicrobial resistance monitoring. These government-led policies are being institutionalized to improve IPC [4].

Meanwhile, multidrug resistance is a growing threat to public health and an emerging crisis [5]. The level of antibiotic usage in South Korea (28 defined daily doses [DDD] per 1000 inhabitants per day [DID]) is still much higher than the average for other Organisation for Economic Co-operation and Development (OECD) countries (18 DID) as of 2012 [6,7]. According to the Korea ASP survey in 2015, the number of infectious disease (ID) specialists increased from previous surveys in 2006 and 2012, but the proportion of hospitals operating ASP did not increase; ASP were limited to top-tier general hospitals [8]. The survey concluded that ASP were not proliferating because of the lack of clinical pharmacists and adequate reimbursement from the national health insurance (NHI) system to support ASP [8]. Pharmacists play crucial roles in IPC/ASP; in fact, they can be facilitators or intermediaries at the opinion-gathering stages [9]. However, few studies have investigated the opinions of ASP stakeholders, such as ID physicians, non-ID physicians, and clinical pharmacists, especially at the level of community health systems (CHS).

The primary aim of this study is to identify the views and potential setbacks to successful ASP implementation at CHS which has limited resources. The secondary aim is to identify priorities of future infection-related service needs.

## 2. Results

### 2.1. Current Perception of ASP Operation

Professionals did not recognize ASP well. Physicians seemed to have a greater awareness of IPC management perspectives apart from ASP *(phy5-1-6, phy7-1-9).* Physicians responded that there were governmental initiatives and investments to improve infection control, but the efforts were not enough to change practice significantly. Moreover, they complained of a lack of institutional support and the overwhelming burden on standard measurement index of hospital accreditation without proper direction and education (*phy2-1-20, phy9-1-10)*.

“In university hospitals and large hospitals, research and education are a big part of the task, but in local community hospitals like us, that function is a bit weak.” *(phy9-1-10).*

Pharmacists’ perception of the ASP was primarily about what the pharmacist should be doing. Pharmacists claimed that they intervened in renal dosing adjustments, durations of antimicrobial use, and prescription evaluation when requested (*pharm2-3-54, pharm2-7-153).*

“I assess renal function with eGFR when the patient’s kidneys are not functioning well.”

“It’s not easy to memorize all of the renal doses, right? So, I, as a pharmacist, always check the doses using the drug information resources to make sure they are adjusted appropriately.” (*pharm2-3-54*).

### 2.2. Current Perception of Barriers and Future Direction of ASP

#### 2.2.1. Conflicts over Primary Responsibility and Scopes of Practice

Antimicrobial selection processes to treat IDs, including those caused by resistant microbial strains, appear to vary among physicians. Pathogen-specific antimicrobial choices based on susceptibility results are common practice among ID physicians in the efforts to prevent the further emergence of resistant pathogens and antimicrobial abuse. On the other hand, non-ID primary teams prioritize following clinical status when prescribing antimicrobials.

“I see many more of my patients with pneumonia were actually infected by pretty nasty resistant strains than patients in the other services... in order to treat them effectively. I have found myself using several antimicrobials instead of narrowing them down.” *(non-IDphy7-2-44)*

“If the strain shows good susceptibility to an antimicrobial, I would choose the antibiotic with a narrow spectrum.” *(IDphy3-3-76).*

These different approaches between the primary team and ID physicians may result in different antimicrobial treatment strategies in practice. The primary team showed hesitance to comply with ID recommendations due to concern of potential misunderstanding patients or the treatment plan of the primary team due to not following patients throughout the hospitalization. During the interviews, ID physicians also expressed that ID consults were sometimes disregarded and were requested late. Unfortunately, this meant that ID consults were often pointless *(phy6-1-20)*. ID physicians believed that their roles were limited in the approval of restricted antimicrobial agents. In hospitals where ASP are not well implemented, the role of an ID physician seems to be limited. Some primary team physicians felt that ID consults (e.g., approval of the use of restricted antimicrobials) undermined their prescribing privileges *(phy4-6-165)*. It seems that intra-professional collaborative communication is not well established, which makes it difficult to reach a consensus about treatment *(phy3-1-14, phy3-7-194, phy5-1-7).*

#### 2.2.2. Antimicrobial-Related Problems (ARP)

“During my consulting (ID) round, I often find patients who have been prescribed incorrect doses; there was a case where one-fifth of the dose should have been prescribed, but the regular dose had been administered instead. There are cases when medications are administered without adjusting the prescriptions despites of changes in organ function.” *(phy3-4-109).*

The most frequently mentioned problem in ARP is the inappropriate doses of antimicrobials. Inappropriate renal dosing adjustments of aminopenicillins, dosing frequency of levofloxacin, and therapeutic drug monitoring of vancomycin were also identified as problems *(phy2-5-123, phy6-3-68)*. There was concern over non-ID primary physicians adjusting doses for patients with decreased renal or hepatic function by ID physicians and pharmacists while overlooking acute changes or comorbidities *(phy2-5-125, phy7-3-66, pharm2-3-63)*. Nevertheless, pharmacists were recognized as the most suitable profession to monitor drug–drug and drug–food interactions *(phy2-5-126)*. Physicians believed that pharmacists are trained to reconcile medication problems with good accessibility to drug information *(phy6-4-100)*. A lack of conversion from parenteral to oral routes was also mentioned, as pharmacists found that parenteral medication often continued until discharge regardless the clinical stability *(pharm1-1-27, pharm1-1-32, pharm2-9-211)*.

#### 2.2.3. Lack of a Financial Support and Reimbursement System

Physicians urged hospitals to invest effectively in infection management. They also complained that the government did not adequately subsidize ASP. The scarce budget and governmental subsidies cannot secure the needed labor force of hospital healthcare professionals and, ultimately, results by not allocating enough manpower dedicated to ASP *(phy1-1-16, phy2-6-154, phy4-9-271, phy8-1-13, phy8-1-22, phy8-2-33, pharm2-3-68, pharm2-6-139)*.

#### 2.2.4. Labor-Intense Medication Preparation and Dispensing Services in Pharmacy

Pharmacists could not afford the time to validate antimicrobial prescriptions before dispensing them due to overwhelming labor-intense pharmacy work and shortages of pharmacists and staffs. Moreover, as the technicians are not allowed to involve in medication preparation by the law on pharmaceuticals in Korea, allocating time for any clinical activity by pharmacists is difficult. Ironically, nurses tend to be involved in antimicrobial management, while pharmacists have no chance to participate *(pharm2-12-269, pharm2-8-210)*.

“We don’t have time to confirm if antibiotic changes are being made correctly. We barely have time to evaluate oral antimicrobial agents for patients.” *(pharm2-9-216)*

#### 2.2.5. Preferred ASP and ARP Services by Gap Analysis

Physicians and pharmacists recognized that most ASP (8 of 10 services) and ARP (8 of 11 services) were not implemented (Figure 1D and Figure 2D) in the present state. Despite the services unavailablity, physicians and pharmacists were willing to utilize the relevant services (Figure 1A’ and Figure 2A’) although they showed different needs of future services (Figure 1B’,C’ and Figure 2B’,C’).

Physicians were unaware of some services at their practice sites (Figure 1C and Figure 2C), on the other hand, pharmacists claimed that they had been mediating those services, such as evaluating antimicrobial regimens, duplications, and interactions (Figure 1C and Figure 2C). Moreover, pharmacists suggested implementing 9 (except 1) ASP- and 11 (all) ARP-related services in the future (Figure 1A’,C’ and Figure 2A’,C’), but only 5 in ASP and 6 in ARP services were preferred by physicians (Figure 1A’ and Figure 2A’).

As an example, “Prospective audit with direct intervention and feedback” of ASP and “Inappropriate medication dose, dosage form, schedule, route of administration, or method of administration” of ARP were mentioned several times during the interviews by both professions. The gap analysis visualized the result in which pharmacists believed that the service has been implemented, but physicians do not (Figure 1C and Figure 2C), and in the future, both physicians and pharmacists agree on the need for this service (Figure 1A’ and Figure 2A’ ).

Awareness of the ASP was similar in both occupations, but physicians’ views in services that they hope to be developed in the future appear to have different priorities than those of pharmacists.

## 3. Discussion

The fact that occupational conflicts between ASP and non-ASP healthcare professionals constitute a barrier to the success of ASP has been mentioned in several studies [10,11,12]. An ASP can be defined as a process that enhances the effectiveness of treatment by sharing the decision-making stage in the treatment of ID among a multidisciplinary team of experts. In this study, conflicts related to scopes of responsibility and interpretation of the treatment guidelines were found not only inter-professionally, between primary care physicians and pharmacists, but also intra-professionally, between ID specialists and primary team physicians. Since antimicrobials articulate important inter-professional asymmetries, pharmacists should be positioned as delimited negotiators within the context of medical prescribing power. A nuanced understanding of the characters of inter-professional negotiations is crucial for improving the use of antimicrobials within and beyond the hospital. Pharmacist participation is the starting point of ASP team building and can facilitate consensus among multidisciplinary healthcare professionals regarding prescription rights. In our study, non-ID specialists, ID specialists, and pharmacists understood the necessity of ASP and demanded the support of a central system and interrelated team action. However, they had drawn their own boundaries within the scopes of practice and the allowed collaborative share. Our in-depth interviews (IDIs) and focus group discussions (FGDs) suggest that the primary responsibility of patient care by the primary team physician should be respected, but at the same time, the antimicrobial treatment decision by the ASP should not be contestable. Pharmacists should also understand the physicians’ decision processes to apply evidence-based pharmacotherapy.

In Korea, ASP-trained clinical pharmacists and medical microbiologists are lacking, and thus a pharmacist or medical microbiologist is not usually a core member of an ASP team, as determined in survey reports conducted in Korea [8,13]. The percentage of pharmacists working at medical institutions is 22.1% and 24.1% in Japan and the US, respectively, but about 11.7% in Korea [14]. In addition, because pharmacy technicians do not operate within the legal framework, job separation and job descriptions of pharmacists are not standardized. It is necessary to reduce the labor-intense preparation work by legalizing the pharmacy assistant personnel system and by upgrading pharmacy automation. Furthermore, there is no financial reimbursement for the professionals who perform and maintain ASP in the healthcare setting in Korea [15]. It might be hard to see a direct correlation between the reimbursements of ASP activities and the proper use of antibiotics. Although the MOHW has reimbursed medical institutions for infection prevention and management measures according to their grades since 2017, this reimbursement policy is limited regarding such measures such as strengthening facilities and human resources standards to prevent infection, improving infectious disease management, expanding the medical surveillance system, and supporting infectious disease management for small and medium medical institutions [4]. Based on this reimbursement policy, Grade 1 sites are allowed one person in charge of IPC per 150 beds, and about $1.7/patient/day is reimbursed. This budget is, in fact, mainly used for hiring nurses dedicated to IPC. Physicians did not seem to clearly distinguish responses to IPC from ASP, and they did not focus solely on responses about ASP for the purposes of this study. It can be seen that the awareness of the medical staff about infection management is reflected in the priority given to infection management and prevention but not to antimicrobial medication management. When answering the questions on current experience with ASP, doctors were more interested in the environment, facilities, and treatment materials to prevent infection than antibiotics.

The interview results of our study are largely in line with existing research on barriers to overcome when introducing ASP. However, the differentiation of this study between other research results lies in its making up for the weaknesses of study methods with FGD and IDI focusing on infectious disease management. Human behaviors are not easily verbalized, especially if they are considered a generally accepted part of the culture. However, human behaviors are easily observed. When responding to our supplementary questions, interviewees reinforced these points. Our findings reveal a weak correlation between the current service establishment and future service requirements (Figure 1 and Figure 2). Gaps between pharmacists and physicians seem to be profound. Even though physicians felt that most of the current ARP-related services were not well established, their demands for the future were less extensive than those of the pharmacists. Particularly in connection with services related to formulary restrictions and preauthorization requirements, physicians appeared to recognize that the current service is excessive. They want pharmacists’ services in the areas of ADRs, drug interactions, and adherence, which occur after they have prescribed antimicrobials. The current lack of services that physicians strongly desire can be prioritized in the further development of ASP. Pharmacists are well aware of the need for all ASP services and the further efforts needed to systematize and promote better services within the system currently in place. 

There are limitations that should be noted. As for the characteristics of research methods, our study has a major limitation which is the small number of interviewees both overall and in subgroups by specialty. Another is the study being centered on a specific region that is in the Seoul capital area. As efforts to overcome such limitations, the interviews were conducted until saturation was achieved among experienced physicians and pharmacists. Three hospitals with potential infrastructure to develop ASP were selected in the Seoul capital area, which is home to more than 48.2% of the national population of South Korea. The results may still not completely reflect the opinions of all physicians and pharmacists at CHS of Korea so further study is warranted. Nonetheless, our results suggest several considerations for the future development of ASP with ARP. Lastly, there might be a bias in performing interviewing, coding and selecting the content. To reduce individual biases, a trained interviewer and researchers with pharmacy backgrounds and knowledge of ASP concepts were involved to assist with coding and theme development, and we additionally analyzed the results using a gap-analysis. The gap analysis was applied to visualize our qualitative analysis; thus, the extrapolation of the result should be carefully interpreted.

We propose the following potential solutions to fix the problems afflicting ASP in CHS, according to our study. The first would be (1) information technology (IT) support. As Korean law allows only pharmacists to process medication preparation and dispensing, which leads to the limited clinical involvement of pharmacists in pharmaceutical care service, effective ASP-specific IT support is necessary to overcome those challenges. Several features should be included such as (a) sorting the list of patients who are on antimicrobials from the electronic medical records (EMR), (b) reviewing antimicrobial orders of patients, (c) checking relevant laboratory results of white blood cell counts and renal/hepatic functions, and (d) reporting microbiology identification with cultures and sensitivity. Lastly, it is critical to have the function of (e) ways of communication among primary providers, ID physicians, and pharmacists. (2) Because CHS is community-based, ASP should be tailored to local characteristics, patient composition, and the demands of hospital professionals. The first step is to prioritize the services. In our interview, although there were some differences among professionals on what they prefer, pharmacists are able to find what service would be more accepted by physicians. Based on our findings, although physicians had different preferences in prioritizing each service item of ASP/ARP, services regarding dosage review, ADR, drug–drug and drug–food interactions were often mentioned first.

The efforts to develop and improve ASP are ongoing in CHS, but it has hardly progressed [16]. Loosely regulated medical referral systems, overcrowded healthcare facilities, a lack of expert resources and infection control infrastructure, a lack of organized leadership for medical crises, and an understanding of ASP were mentioned as major problems that hinder ASP in CHS. It should prompt the government to reform the healthcare system and to further invest in manpower and financial compensation [17].

## 4. Methods

### 4.1. Study Design

#### 4.1.1. Setting, Participants, and Interviews

The method of consensual qualitative research was applied in part using a cross-sectional and exploratory research approach of IDIs and FGDs [18,19]. At the end of each session, supplementary questions with yes/no dichotomous responses were posed to quantitatively assess the needs for ID-related services. 

We targeted three CHSs, each with between 600 and 850 beds. The sites had teaching capacity but minimal interdisciplinary team activity. Physicians working full-time, certified by the board of internal medicine, and those holding antimicrobial prescription privileges were interviewed individually because of their different roles and specialties in practice as primary vs. non-primary and ID vs. non-ID. FGD was applied to pharmacists working full-time, having at least 3 years of hospital experience. Their major responsibility was mostly the preparation and dispensing of medications (Table 1). Participants were purposely selected via a snowball sampling strategy. IDI and FGD arrangements were made in advance by email or phone conversation. Written informed consent was obtained from each participant on the day of their respective interviews, and confidentiality was maintained of all personal information. The interviewer was a graduate student with experience conducting semi-structured interviews. To ensure interview quality, a second person supervised each interview. Interviews ranged between 40 and 60 min in length. The audio-recorded interviews were transcribed verbatim by a paid undergraduate student experienced in transcription. Once completed, all of the transcripts were checked against the audio recordings to ensure transcription accuracy. Recruitment was finalized when data saturation was reached [20]. 

#### 4.1.2. Research Team

The research team included four researchers: a graduate student (PharmD), a clinical pharmacist, a clinical pharmacy faculty member (PharmD, PhD), and a pharmacy director (PhD). The pharmacy director served as an auditor for the study and did not participate in team meetings, which allowed them to provide an independent opinion throughout the audit process. All team members had previous experience working in secondary general hospitals and working on at least one qualitative study. The researchers discussed potential problems throughout the research process to ensure the integrity of the data. 

#### 4.1.3. Development of the Interview Topic Guide and Supplementary Questions

To facilitate effective interviews, a semi-structured topic guide was developed by the research team and three clinical pharmacy graduate students. They met weekly between March and June 2017 to discuss the search strategy, pool of potential interviewees, target hospital capacity, and interview methods. A literature search was conducted using the following keywords: “antimicrobial stewardship program,” “pharmacist,” “intervention,” “antimicrobials,” “infectious disease,” “satisfaction,” and “demands.” The ASP guidelines published by the Infectious Diseases Society of America (IDSA) and American Society of Health-System Pharmacists (ASHP) were reviewed to incorporate the most up-to-date recommendations for performing hospital infection control and pharmacy collaborative activity into the topic guide [21,22]. The relevant literature was summarized into the following categories: issues to be resolved, current challenges, the items of ASP-based services, member composition, and the types of pharmacy intervention. Once the interview topics were drafted, they were modified further by consultations with experts and subsequent discussions involving the research team and the pharmacy students. The sections included in the final topic guide were “perceptions and potential setbacks of current ASP” and “suggestions and future demands for ID-related services”. Supplementary questions were prepared in two parts. The first part had 10 questions asking about previous and current experiences of ASP and future ASP-related needs. The second part had 11 questions about previous experience and future needs related to ARPs. The interviewees were prompted to respond “yes” or “no” to each question (Table 2) [20,21].

### 4.2. Data Analysis

Each line of the interview transcripts was numbered consecutively. Data were thematically coded and analyzed using a framework approach in Microsoft Excel 2017 (Microsoft Corp., Redmond, WA, USA). Research team members were involved in the processes of indexing, charting, and mapping the identified topics, which were then confirmed by a clinical faculty member who checked the coding scheme and each transcript for coding errors. Where representative quotations were presented in this report, references after the quotation indicate the occupation of each participant and the paragraph and line numbers (e.g., “phy1-2-16” refers to the physician coded 1, paragraph 2, line 16) of the transcript. The transcript is publically available at https://github.com/SandyRhie/FGI.

Gap plots were generated to quantify the gaps between physicians and pharmacists in terms of their awareness of the availability and demands of the services that they think should be implemented in future ID management. The level of service provision was considered well established if more than half of the respondents (cutoff: ≥6/11 physicians or ≥6/11 pharmacists) responded “yes”; the same criterion applied to deem future services “necessary.” 

The study was approved by the Institutional Review Board of Ewha Womans University (IRB No. 143-7).

## 5. Conclusions

Our findings emphasize the collaborative relationships between experts and strategies to build a multidisciplinary working environment to develop ASP in the CHS with limited resources and support. Based on the results of the gap analysis, there are some points that need to be changed. Such a communication-promoting interdisciplinary activity is being piloted by our research team in a community-based hospital. We designed the ASP EMR to document any relevant notes in the one section no matter which healthcare professional is involved. We further set up the co-signature function by ID physician after the pharmacy consultation note has been left. It is believed to increase the acceptance of the ASP intervention notes by the primary team. Pharmacists should be available to fulfill the important role of mediating and regulating opinions among physicians and other team members.

The other solution would be reimbursement. It motivates the effort to implement ASP services and rapport in a collaborative working environment. A post-cost-avoidance analysis would be warranted in the near future.

## Figures and Tables

**Figure 1 antibiotics-08-00252-f001:**
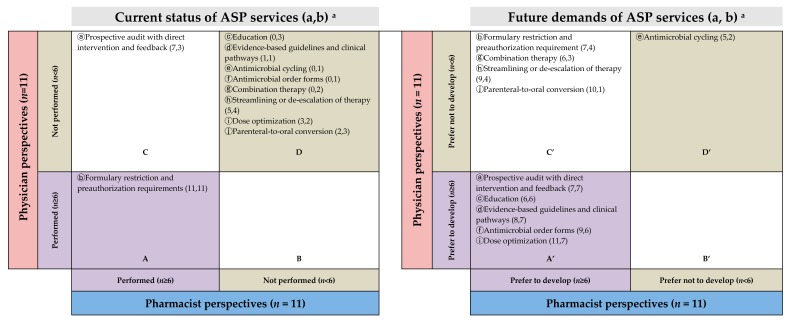
Gap analysis of the current status (left) and future preferred ASP services (right). ^a^ (a,b) means (number of pharmacists who answered “yes,” number of physicians who answered “yes”). ASP: antimicrobials stewardship programs.

**Figure 2 antibiotics-08-00252-f002:**
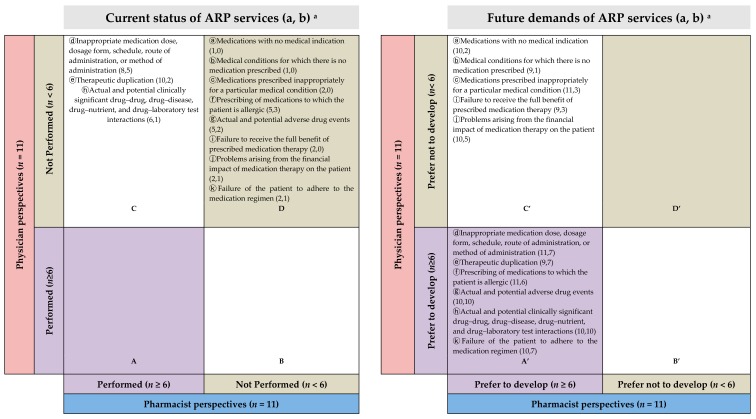
Gap analysis of the current situation (left) and future preferred ARP services that suggest pharmacists’ interventions (right). (a, b) means (the number of pharmacists who answered “yes”, the number of physicians who answered “yes”). ARP: antimicrobials-related problems.

**Table 1 antibiotics-08-00252-t001:** Interviewees’ baseline characteristics.

Characteristics		Physicians *N* = 11	Pharmacists *N* = 11
Gender	Male	7	0
	Female	4	11
Age (years)	26–35	1	5
	36–45	7	4
	46–55	3	2
Work experience (years)	≤5	1	0
	6–10	3	6
	11–15	4	3
	16–20	2	2
	≥21	1	0
Employment	Community-based hospital	11	11
Medical subspecialty	Infection	5	NA
	Pulmonary *	4	NA
	Hematology	1	NA
	Cardiology	1	NA

N/A, not applicable. * Two of pulmonologists are in charge of intensive care.

**Table 2 antibiotics-08-00252-t002:** Questions in the gap analysis about ASP and ARP.

Services	Is the Service Currently Performed? Answer Yes/No	Is the Service Preferred to Develop or Improve in the Future? Answer Yes/No
**Part 1. Services in the ASP**		
a. Prospective audit with direct intervention and feedback	□ Yes □ No	□ Yes □ No
b. Formulary restriction and preauthorization requirements	□ Yes □ No	□ Yes □ No
c. Education	□ Yes □ No	□ Yes □ No
d. Evidence-based guidelines and clinical pathways	□ Yes □ No	□ Yes □ No
e. Antimicrobial cycling	□ Yes □ No	□ Yes □ No
f. Antimicrobial order forms	□ Yes □ No	□ Yes □ No
g. Combination therapy	□ Yes □ No	□ Yes □ No
h. Streamlining or de-escalation of therapy	□ Yes □ No	□ Yes □ No
i. Dose optimization	□ Yes □ No	□ Yes □ No
j. Parenteral-to-oral conversion	□ Yes □ No	□ Yes □ No
**Part 2. ARPs that suggest pharmacist’s intervention**		
a. Medications with no medical indication	□ Yes □ No	□ Yes □ No
b. Medical conditions for which there is no medication prescribed	□ Yes □ No	□ Yes □ No
c. Medications prescribed inappropriately for a particular medical condition	□ Yes □ No	□ Yes □ No
d. Inappropriate medication dose, dosage form, schedule, route of administration, or method of administration	□ Yes □ No	□ Yes □ No
e. Therapeutic duplication	□ Yes □ No	□ Yes □ No
f. Prescribing of medications to which the patient is allergic	□ Yes □ No	□ Yes □ No
g. Actual and potential adverse drug events	□ Yes □ No	□ Yes □ No
h. Actual and potential clinically significant drug–drug, drug–disease, drug–nutrient, and drug–laboratory test interaction	□ Yes □ No	□ Yes □ No
i. Failure to receive the full benefit of prescribed medication therapy	□ Yes □ No	□ Yes □ No
j. Problems arising from the financial impact of medication therapy on the patient	□ Yes □ No	□ Yes □ No
k. Failure of the patient to adhere to the medication regimen	□ Yes □ No	□ Yes □ No

IPC/ASP, infection prevention, and control programs/antimicrobial stewardship programs; ARP, antimicrobial-related problems.

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
