# Peer review of "Antimicrobial Stewardship Programs in Community Health Systems Perceived by Physicians and Pharmacists: A Qualitative Study with Gap Analysis"

_antibiotics, 2019, doi:10.3390/antibiotics8040252_

Round 1

Reviewer 1 Report

The authors have prepared a gap analysis relating inter- and intra-professional opinions on the current and future needs for hospitals with respect to antimicrobial stewardship programs. While the authors present the gathered information succinctly, the material is not necessarily novel. The current challenges and problems associated with ASPs globally are universally present and generally accepted as needing addressed and ameliorated. Although the authors provide the sentimental reflections of pharmacists and physicians, the manuscript lacks any offer of how to remedy the current situation. What seems to be missing are suggestions for the implementation of measurable and actionable activities to rectify the deficits in many ASPs.

Overall, this reviewer has a major concern that nothing is included addressing how to actually fix the problems afflicting ASPs. It is acceptable to bring additional awareness to the issue, but the manuscript seems incomplete without provisions for the solution to the issue.

Of minor concern: Line 271 - It is unclear what the authors are trying to express in the sentence, "Under the limited resources in CHS, ASP in CHS should approached by specific to the hospital."

Reviewer 2 Report

The study describes perceptions of current Antimicrobial stewardship programs, potential setbacks of successful ASP implementation, and future demands on ASP services among physicians and pharmacists in community health system.

General comments-

Abstract: Give 1-2 lines more on the Antimicrobial stewardship programs (ASPs) and can reduce redundancies in describing the setbacks identified for ASP improvements (Lines 36-43).

Introductions: Introduction is well written and concise.

Results: Authors have cited their physicians as “phy5-1-6, phy7-1-19” (e.g. -Line 76, 80) or pharmacist as “pharm2-2-48(line 83). There are several such examples, authors should mention to refer to methods section “data analysis” for annotations of these physicians or pharmacists. A link for the transcript should be provided.The subheading 2.2.2. Antimicrobial-Related Problems (ARP) does not justify the results provided, this should be changed.

Discussion: IDI, FGD and MOHW annotation needs to be mentioned. Table 1 is not clear, needs to be re-written.

References: Link for reference 4, 18 do not work.

Author Response

Response to Reviewer 2 Comments

Point 1: General comments-

Abstract: Give 1-2 lines more on the Antimicrobial stewardship programs (ASPs) and can reduce redundancies in describing the setbacks identified for ASP improvements (Lines 36-43).

Response 1: Thank you for your comment and suggestion. I agree that it would make abstract more clear. Thus, I corrected the abstract including points that you suggested. I also reduced redundancies in describing the setbacks identified for ASP improvements (Lines 36-39).

Add) Antimicrobial stewardship programs (ASPs) is one of the most important strategies for managing infectious disease treatment and preventing antimicrobial resistance. (Line 28, highlighted in yellow)

Rephrase redundancies) Identified setbacks were unorganized institutional leadership, the undefined roles of healthcare professionals, a lack of reimbursement, the hierarchical structure of the health system, and the labor-intensive working environment of pharmacy services. (Line 36-39)

Point 2: Results: Authors have cited their physicians as “phy5-1-6, phy7-1-19” (e.g. -Line 76, 80) or pharmacist as “pharm2-2-48(line 83). There are several such examples, authors should mention to refer to methods section “data analysis” for annotations of these physicians or pharmacists.

Response 2: Thank you for your comment. It was mentioned but I checked to make sure these annotations should show in data analysis of methods section. (Line 326-328)

~, references after the quotation indicate the occupation of each participant and the paragraph and line numbers (e.g., “phy1-2-16” refers to the physician coded 1, paragraph 2, line 16) of the transcript.

Point 3: A link for the transcript should be provided.

Response 3: Thank you for your suggestion and I upload the transcripts (in Korean) in Github repository. The link is publically accessed to https://github.com/Sand. It was mentioned at Data analysis of the method section (Line 328, yellow highlighted)

The transcript is publically available at https://github.com/SandyRhie/FGI

Point 4: The subheading 2.2.2. Antimicrobial-Related Problems (ARP) does not justify the results provided, this should be changed.

Response 4: Thank you for your concern and suggestion. My intention was to express the interview results mentioned at 2.2.2. Antimicrobial-Related Problems (ARP) connecting with the result of a gap analysis. However, I understand your points, so that I made a linkage sentences and deleted sentences about aseptic technique.

Add) The most frequently mentioned in ARP was the inappropriate dose of antimicrobials (Line 120, highlighted in yellow)

Delete) The incompliance of aseptic process of handling injectable drugs in the wards that were not fully aseptically equipped and lacked trained staffs were bought up,…

Point 5: Discussion: IDI, FGD and MOHW annotation needs to be mentioned.

Response 5: Thank you for your comments. I make sure MOHW annotation is mentioned in the Introduction section (Line 52, highlighted in yellow). In addition, IDI and FGD annotation were added in the discussion section (Line 188, and 189, highlighted in yellow).

Point 6: Table 1 is not clear, needs to be re-written.

Response 6: Thank you for your comment. During the revision process, I found that Table 1 is not relevant to the study goals and content so that it was deleted.

Point 7: References: Link for reference 4, 18 do not work.

Response 7: Thank you for your corrections. I changed reference 4 and 18 into reference (Line 372-374 and 409-410 each, highlighted in yellow).

4.        Kwon, K.T.; Lee, W.K.; Yu, M.H.; Park, H.J.; Lee, K.H.; Chae, H.J. The impact of infection control cost reimbursement policy on trends in central line-associated blood stream infections. Open Forum Infect. Dis. 2018, 5(Suppl.1), S613.

18.     Hill, C.E; Knox, S.; Thompson, B.J.; Williams, E.N.; Hess, S.A.; Ladany, N. Consensual qualitative research: An update. J. Couns. Psychol. 2005, 52, 196–205.

Reviewer 3 Report

In the manuscript entitled "Similar demands with different perceptions of antimicrobial stewardship programs in community health systems by professionals: a qualitative study with gap analysis", Park and colleagues describe the set up of a study conducted to qualitatively evaluate the perception of antimicrobial stewardship programs by professionals from Korean community health systems. A total of 8 physicians and 11 pharmacists were enrolled in the study. From the formal point of view the manuscript is well organized and the English usage is adequate. In my view, the work presents several limitations, including the very limited number of participants which belong to 3 distinct hospital settings.  The Results section is in part dedicated to transcribe selected opinions from the participants, offering a qualitative view form a few professionals working in 3 different hospitals from one single country, thus reflecting the specific socio-economical and political status of a specific community. Therefore, conclusions from the very small universe of professionals opinions cannot be generalized and the interest is quite limited. Below follows some minor issues:

Figure 1, left panel, C: the sentence is confusing: "(Performed inadequately by pharmacists´perspectives, but NOT by physicians)", is it OK?

Figure 2, left panel C : same comment

line 271: Re-write the sentence, it is quite confusing.

line 289: ", having at least ..."

line 407-410: is it OK?

Reviewer 4 Report

I read with interest the qualitative study (based on IDI and FDG) by Park and colleagues. Overall, the writing is easy to follow. Below are some comments on the technical merits of the paper.

Major comments

Although I understand that the achievement of saturation partly allowed for a reduced sample size, it is also true that there are subgroups that may not be representative, for example only 4 ID physicians may not guarantee a saturated view of ID specialists. This could be even more important for non-ID physicians. There were just one pulmonologist and three hematologists. Several other specialists (e.g., internal medicine, intensive care) were not represented There is an imbalance as regards the distribution of interviewees in CHS hospitals. According to tables, pharmacists were from 3 hospitals, physicians from 2.

Minor comments

“The number of antimicrobials prescribed increased steadily from 26.9 defined daily doses (DDD) in 2008 to 31.7 DDD in 2014 but dropped to 31.5 DDD in 2015 [6]”. The numbers in the cited publication seems different, please check. They are also probably not DDD, there is a lack of denominator.

Round 2

Reviewer 1 Report

The submitted manuscript appears to have been revised sufficiently for publication.

Reviewer 2 Report

The comments were addressed, hence no further comments.

Reviewer 3 Report

The major concerns, related to the small number of professionals and the biased results due to the fact that  the study is centered on a specific region reflecting a singular situation still remain. Although I recognize that the authors have made an effort to improve the quality of their study, this opinion study is quite limited and biased.

Reviewer 4 Report

Thank you for your kind responses to comments. Just a last minor comment. I am still not convinced that reaching saturation allows for generalization in this complex setting. I do think that in the discussion it should be clearly and firstly reported that a very major limitation is the limited number of interviewees, which remains very small, both overall and (especially) in subgroups by specialty.
